# C-Reactive Protein Is Associated with Physical Fitness in Breast Cancer Survivors

**DOI:** 10.3390/jcm12010065

**Published:** 2022-12-21

**Authors:** María Romero-Elías, Alejandro Álvarez-Bustos, Blanca Cantos, Constanza Maximiano, Miriam Méndez, Marta Méndez, Cristina G. de Pedro, Silvia Rosado-García, Antonio J. Sanchez-Lopez, David García-González, Héctor Cebolla-Boado, Ana Ruiz-Casado

**Affiliations:** 1Department of Sports Sciences, Sport Research Centre, Miguel Hernandez University of Elche, 03202 Elche, Spain; 2Biomedical Research Center Network for Frailty and Healthy Ageing (CIBERFES), Institute of Health Carlos III, 28029 Madrid, Spain; 3Department of Medical Oncology, Hospital Universitario Puerta de Hierro Majadahonda, IDIPHISA, 28222 Madrid, Spain; 4Biobank, Puerta de Hierro-Segovia de Arana Health Research Institute, 28222 Madrid, Spain; 5Faculty of Medicine, Autonomous University of Madrid, 28029 Madrid, Spain; 6Spanish National Research Council (CSIC), 28006 Madrid, Spain

**Keywords:** biomarkers, cardiorespiratory fitness, physical condition, breast cancer, physical activity

## Abstract

Background: Physical fitness (PF) is an expression of the physiological functioning of multiple body components. PF is an important prognostic factor in terms of cardiovascular mortality, cancer mortality, and all-cause mortality. PF has been related to some biomarkers in the general population but not in breast cancer survivors (BCS). Purpose: To evaluate the effects of PF on biomarkers potentially related to physical activity (PA) in a sample of BCS. Methods: Cross-sectional study. A total of 84 BCS (mean age 54) who had finished their treatment were recruited. Different components of PF were evaluated, namely body composition (anthropometry), cardiorespiratory fitness (one-mile walk test), muscular (handgrip and sit-to-stand timed test), and motor (gait speed) components. Sexual hormones, inflammation, and insulin resistance biomarkers were measured. Results: C-Reactive Protein (CRP) was associated with every component of physical fitness: cardiorespiratory fitness (*p*-value = 0.002), muscular (sit-to-stand timed test, *p*-value = 0.002) and motor (gait speed, *p*-value = 0.004) components, and body composition (body mass index, *p*-value = 0.003; waist, *p*-value < 0.000; and waist-to-hip index, *p*-value = 0.012). CRP also was associated with “poor physical condition,” a constructed variable that encompasses all components of physical fitness (*p*-value < 0.001). Insulin was associated with cardiorespiratory fitness and gait speed (*p*-values = 0.002 and 0.024, respectively). Insulin-like Growth Factor-1 was negatively associated with waist perimeter and waist-to-hip ratio. Conclusions: CRP can also be considered an indicator of poor PF in BCS. Implications for cancer survivors: in case of elevation of CRP indicating cardiovascular risk, health professionals should recommend lifestyle changes to improve BCS physical condition.

## 1. Introduction

Assessment of physical performance or physical fitness (PF) is of utmost interest to physicians and specifically to medical oncologists, as it is highly informative for the process of therapeutic decision-making. PF was defined by the American Academy of Physical Education as “the ability to carry out daily tasks with vigour and alertness, without undue fatigue and with ample energy to engage in leisure-time pursuits and to meet the above-average physical stresses encountered in emergency situations” [1]. Nowadays, performance- and health-related PF are distinguished; performance-related fitness refers to the abilities associated with athletic performance, and health-related fitness is associated with the risk of the development of diseases associated with physical inactivity [2]. The concept of health-related PF includes different components: (a) body composition, (b) cardiorespiratory, (c) muscular (muscular strength and endurance), and (d) motor components (balance, flexibility, speed, etc.) [3].

Cardiorespiratory fitness (CRF) is one of the most widely studied components of PF. In fact, CRF has been claimed to be a vital sign incorporated in cardiovascular (CV) risk assessment because of its usefulness and simplicity [4], but especially due to its strong association with cardiovascular (CV) disease, mortality, and all-cause mortality [4]. Higher CRF is also associated with survival in cancer patients [5]. Muscular strength has also been associated with CV-related mortality [6], cancer mortality [7], and all-cause mortality [8]. In addition, obesity can increase CV morbidity and mortality [9].

Studying risk factors for CV disease in breast cancer survivors (BCS) is relevant because, thanks to the high rates of success in treating BCS and due to the consequences of the treatments, there has been an increase in CV disease and nowadays is becoming the most important cause of death among middle-aged BCS [10]. Identifying people at risk of developing CV disease and implementing effective measures of primary prevention, such as a change in their lifestyle, is essential in the surveillance of BCS. Physical activity (PA) is associated with reduced mortality in breast cancer, but the biological pathway that regulates this association remains unclear [11]. The purpose of this report is to evaluate the effects of health-related PF on biomarkers that are potentially related to PA in a population of BCS who had finished their treatment.

## 2. Material and Methods

### 2.1. Participants

Participants were recruited at the Department of Medical Oncology of the Hospital Universitario Puerta de Hierro Majadahondafrom 2017 to 2020. Inclusion criteria were being older than 18, diagnosed with a localized diagnosis of breast cancer (stages I–III), having finished their treatments (trastuzumab and hormonal therapy were allowed during the evaluation), being disease-free, and having signed the written consent form.

### 2.2. Study Design

A descriptive, prospective, and cross-sectional design was chosen to identify the associations between biomarkers and PF variables. The protocol entailed PF tests, anthropometric measures, blood analysis, and socio-demographic data. Clinical data were obtained from clinical records. The study protocol was approved by the pertinent Ethics Committee and was consequently performed in agreement with the Declaration of Helsinki and relevant European law.

### 2.3. Measures

**Anthropometry.** Body mass index (BMI) was determined as weight/height squared (kg/m^2^). Waist and hip circumference were measured following the international recommendations, and waist-to-hip index (WHI) was calculated.

**Physical fitness.** CRF was assessed using the one-mile walk or the Rockport test. All tests were performed at the hospital and timed using a stopwatch. Participants’ VO_2MAX_ was calculated according to age, sex, and body mass-specific equations detailed elsewhere [12]. Heart rate was measured before and immediately after the tests with a heart rate telemeter (Xtrainer Plus; Polar Electro OY, Kempele, Finland), and the recovery heart rate was measured one minute after finishing. The time required to complete the distance (1606 m) measured the gait speed.

Regarding muscular strength, handgrip strength was measured using a dynamometer (TKK 5001 Grip-D; Takei, Tokyo, Japan) and scores were recorded in kilograms (to the nearest 0.1 kg). The sit-to-stand (STS) timed test is a test of muscular strength and measures the time (in seconds) required to perform five consecutive repetitions of sitting down and rising from a chair, and it was used to assess lower-limb strength. Participants began the test with their arms crossed over their chest and sitting with their backs against the chair. They were instructed to perform the task “as fast as possible,” starting and finishing in the sitting position.

**Laboratory measurements.** Venous blood was collected from each subject after a 12 h fast and used for the assay of glucose, C-Reactive Protein (CRP) (using the Advia XPT automated biochemistry analyzer), cortisol, estrogens, progesterone, testosterone (using the Advia Centaur XPT Immunoassay System), androsterone, insulin, Insulin-like Growth Factor-1 (IGF-1), and Insulin-like Growth Factor Binding Protein 3 (IGFBP3) (using the Immulite 2000 chemoluminescence system).

Samples and data from patients included in this study were provided by Biobanco Hospital Universitario Puerta de Hierro Majadahonda (HUPHM)/Instituto De Investigación Sanitaria Puerta de Hierro-Segovia de Arana (IDIPHISA) (PT17/0015/0020 in the Spanish National Biobanks Network), and they were processed following standard operating procedures with the appropriate approval of the Ethics and Scientific Committees.

**Socio-economic characteristics.** A self-report questionnaire was used to obtain information about education level, employment status, global annual income, having or not having a partner, disabled family member under care, children under 15 years of age, perceived health, and smoking habits.

**Clinical data.** Clinical data were collected by health professionals from clinical records.

### 2.4. Statistical Analysis

Basic descriptive variables were calculated. All the variables (namely VO_2MAX_, STS, the time spent on the one-mile walk test, handgrip strength, BMI, waist and hip perimeter, and recovery and basal heart rate) used in this analysis are part of a general scope dimension that could be tentatively called Integrated Physical Fitness (IPF). Indeed, it could be argued that all of these variables could be understood as partial or imperfect proxies of IPF. This conceptualization not only facilitates the interpretation of results but also could contribute to a more parsimonious modelization of our effects. A factorial analysis was performed to reduce all of these variables to a single and synthetic PF index. As expected, our variables are reducible to a single dimension to which they jointly contribute. The selected index has an Eigen-value of 3.80, and the proportion of variance accounted for by the factor is 0.71. All of the involved variables substantially contribute to the synthetic factor according to the factor loadings. It can be seen from Table 1 that our index importantly correlates with most of the variables, with STS, handgrip, and heart rate showing the lowest association with this general PF dimension.

Bivariate linear regressions were used to measure the impacts of biomarkers and PF variables. For the sake of simplicity, only β_1_ coefficients with their corresponding standard errors are provided together with the associated *p*-values. All data analyses were conducted using STATA v 15.1 (StataCorp, LLC; College Station, TX, USA) statistical software.

## 3. Results

### 3.1. Descriptive Data

A total of 84 BCS were included in this study. Table 2 presents the following descriptive data: clinical, socio-demographic, anthropometric, and PF-related data. Biomarkers are also described. The mean age of the participants was 51 years, and the mean time since diagnosis was around three years. Most patients were diagnosed at Stages I and II. Hormonotherapy was the most common treatment (83%), and 74% received chemotherapy. Patients were slightly overweight (BMI over 25) with central adiposity values sorted as normal (waist below 88 cm and WHI below 0.85). The CRF assessed through the one-mile walk test (gait speed: 1.7 m/s) was slightly below the recommended value (>28 mL/kg/min for women of 50 years).

### 3.2. Regression Results

The statistically significant regression results are presented in Table 3. All the biomarkers, namely CRP, NLR, cortisol, estrogens, progesterone, testosterone, androsterone, insulin, IGF-1 and IGFBP3, were evaluated for each component of PF. In order to facilitate reading, only significant results were included.

Biomarkers and cardiorespiratory condition. CRP and insulin were negative and significatively associated with the estimated VO_2MAX_ (−0.59, *p* = 0.002 and −0.07, *p* = 0.014, respectively). Insulin levels were associated with basal heart rate (0.13, *p* = 0.006), and testosterone was associated with better recovery heart rate (0.84, *p* = 0.001).

Biomarkers and muscular component. CRP was associated with lower-limb muscle strength (0.19, *p* = 0.002).

Biomarkers and motor components. CRP and insulin were positively and significatively associated with the time required to walk a mile (0.13, *p* = 0.004 and 0.01, *p* = 0.024, respectively).

Biomarkers and body composition. CRP was positive and significantly associated with BMI (0.55, *p* = 0.000), hip circumference (0.71, *p* = 0.003), waist circumference (1.04, *p* = 0.000), and WHI (0.00, *p* = 0.012). IGF-1 was associated with a healthier waist (−0.05, *p* = 0.040) and WHI (−0.00, *p* = 0.049).

Biomarkers and IPF. Only CRP showed a significant positive association with IPF (0.09, *p* < 0.001).

No associations were found between handgrip (muscular strength) and biomarkers, nor were any significant associations found between the rest of the PF variables and biomarkers.

## 4. Discussion

The primary finding of this cross-sectional study was an inverse association between CRP levels and different parameters representative of PF in a population of BCS who had finished their treatments 2–3 years before. CRP is a well-known inflammatory biomarker produced by the liver and atherosclerotic plaques and is a clinical marker for different diseases, such as ischemic stroke, coronary artery disease, hypertension, insulin resistance, peripheral artery disease, and metabolic syndrome [13]. There seems to be a linear association between CRP and cardiovascular disease, with an increased risk of 18% per 1 mg/L increase in CRP level [14].

CRP has already been proposed as a correlate for CRF in the general population [15]. The association of CRP with other components of PF revealed by this study not only strengthens the previous finding but provides us with a more comprehensive perspective that could be very useful in the surveillance of BCS. On the other hand, it has been suggested that CRP levels could be monitored to assess the improvement in CV disease [13] and, if required, to perform interventions that are effective in lowering CRP levels, such as exercise, weight loss, smoking cessation, or diabetes control. In this sample, the mean level of CRP was 2.2 (mg/L), which can be considered a risk value for developing CV disease [16].

The impact of body composition on breast cancer prognosis is very well known. Body composition can be assessed in various sophisticated ways, such as Dual-energy X-ray Absorptiometry (DeXA), plethysmography, total body potassium, and bioimpedance. As BMI is a cheap, simple, and accessible method, it is the basis for the World Health Organization’s (WHO) definition of overweight. Regarding fat distribution, waist and hip perimeters are also very useful. These simple measures are easily accessible and are comparable with other previously published results. In our series, all the variables related to body composition (BMI, hip, waist, and WHI) were positively associated with CRP. It is also interesting to note that BMI and waist perimeter had an important weight in our composite variable (IPF). Weight loss multimodal (diet, exercise, and psychosocial support) interventions for overweight and obese BCS are effective in terms of decreasing body weight, BMI, and waist circumference. Waist and WHI were associated with IGF-1 but not with other markers of insulin resistance.

An important mechanism for the chronic inflammation associated with metabolic syndrome appears to be inflammatory cell infiltration with increased production of inflammatory cytokines in adipose tissue. In addition, metabolic abnormalities associated with insulin resistance (hyperglycemia or elevated free fatty acids) or insulin resistance per se led to cellular inflammatory responses. Thus, the inverse relationship between PF and CRP may reflect a fitness-induced increase in insulin sensitivity [17].

CRF is the ability of circulatory, respiratory, and muscular systems to supply oxygen during exercise. A good CRF involves numerous benefits for health outcomes, such as depression, metabolic syndrome, CV, cancer and all-cause mortality [18]. In the laboratory, CRF is assessed through maximal incremental exercise testing [2]. However, in the clinical setting, the so-called field tests are used; subjects are required to walk briskly on the ground level, and the test renders an estimated VO_2MAX_. Though it is closely and specifically related to PA and leisure-time PA, CRF is a trait, whereas PA is a behavior. CRF is genetically determined, being higher in males and young people, and it improves with the practice of exercise. It also depends on habits, such as smoking or alcohol consumption, body composition, educational status, and the residential-built environment. Zeiher et al., in their systematic review, added CRP as a correlate for CRF [15]. Previously, CRF levels had been described as inversely associated with CRP values in the healthy general population [19], men [20] and cancer survivors [16]. In our sample, VO_2MAX_ was associated with CRP, so increasing leisure-time PA and implementing exercise programs focused on improving CRF is essential in reducing CRP and, therefore, the risk of CV and mortality in cancer survivors. It should be kept in mind that only intensive and prolonged exercise can significantly improve CRF in BCS [21].

Regarding the muscular component, two tests were performed. The handgrip test evaluates the strength of upper limbs, with poorer values of muscular strength being associated with breast cancer, all-cancer, and all-cause mortality, among other health outcomes [22]. Although higher levels of circulating inflammatory factors are associated with lower muscular strength in the general population [23], surprisingly, handgrip strength was the only variable related to PF that was not associated with CRP in our study. Nevertheless, we found associations between CRP and muscular strength of the lower limbs, as measured by the STS test. The relationship between higher CRP and lower performance in the STS test has been previously reported in other studies [24,25], but to our knowledge, not in BCS. There appears to be enough evidence to conclude that higher systemic inflammation is associated with lower muscular strength. Additionally, low muscle mass has been proposed to induce low-grade systemic inflammation directly [26]. In any case, resistance training (a form of PA that is designed to improve muscular fitness by exercising a muscle or a muscle group against external resistance) has been shown to decrease CRP levels in BCS [27]. Therefore, recommending BCS to adhere to a comprehensive program of physical exercise, including resistance training that could improve muscle mass and strength, would be more than warranted in the context of CRP elevation during surveillance.

The fourth component of PF is the motor component, which includes balance and speed, but also flexibility and agility. BCS could have both gait and balance impairments compared with normative values [28]; however, they are rarely measured outside of geriatric assessment. Gait speed has also been recognized as a prognostic factor in terms of mortality in general [29], older, and disease populations and is one of the components of the frailty syndrome [30]. Gait speed was the only variable in our study assessing the motor component, and it was associated with both CRP and insulin. There is some evidence connecting these biomarkers to gait speed in the general population [31,32]. Kuo et al. declared that, regardless of chronic diseases, elevated CRP was associated with multiple domains of disability concerning strength and gait speed [31]. No similar associations have been previously described for BCS, but they might not have been properly studied [33].

However, it is evident that PF and the ability to carry out an active life do not depend on just one of the components but on all of them conjointly. That is why we took advantage of the statistical association of positive results in different tests and designed a variable that comprised all of them, calling it IPF. Interestingly, the association of IPF with CRP was positive, but IPF was not associated with insulin levels or the rest of the biomarkers. This fact reinforces the idea that CRP constitutes a good marker of health-related PF and could be used in the clinical setting. Therefore, we hypothesize that CRF could play a role by inducing an anti-inflammatory status, and this would be mainly and firstly mediated by body composition because all of the related variables were associated with CRP and also because they were the major contributions to the newly constructed variable, IPF.

There is another reason to conjecture that the link between CRP and PF is through body composition. Phase angle, a parameter derived from bioimpedance, is the only other objective and measurable variable that we found to be related to all the domains of PF [34]. It has also been considered a marker of BCS’ health status and functional capacity [34] and has been associated with CRP in the older population [35]. We are not aware of similar studies in BCS, and given the extraordinary importance of CV risk for this population of women, further research is warranted. These results should be tested in intervention trials.

High levels of CRP (>2) can be considered a trigger for the health practitioner to strongly recommend a change of habits in BCS [19]. Some practical implications for healthcare and exercise professionals would include helping BCS to manage their weight. Regarding PA recommendations, they should probably focus on vigorous PA, trying to achieve a better CRF, and systematic resistance muscular training, producing muscular gain. This approach may reduce CRP values, thereby lowering the risk of adverse health outcomes in this population.

The main limitation of our study is its cross-sectional design, which limits the cause–effect association between CRP and PF. A longitudinal design would also allow studying the influence of health-related fitness and CRP outcomes, such as CV and all-cause mortality. Another important limitation is that our study does not address the relationship between changes in individual fitness levels over time and inflammatory markers. Regarding their assessments, the use of a submaximal test for evaluating the cardiorespiratory condition is not ideal. On the other hand, balance and flexibility were not evaluated. Information about processes that could affect PCR levels, such as infectious processes at the time of extracting the samples, was not collected, and this could have biased the results. Information about concomitant medication was not recorded.

Despite the above-mentioned limitations of the study, this work has a large number of strengths: the prospective (though non-longitudinal) design, the objective evaluation of all PF components, and anthropometry. The clinical data were obtained from clinical records, and they were consequently not self-reported. The novelty of the study is another aspect to highlight, as, to date, there is limited scientific evidence on the relationship between health-related PF and biomarkers in BCS.

In conclusion, CRP could be considered a marker of PF in BCS and, consequently, can be used to measurably and pragmatically follow CV risk in this specific population. Acknowledgements: The authors wish to thank the donors, and the Biobanco Hospital Universitario Puerta de Hierro Majadahonda (HUPHM)/Instituto De Investigación Sanitaria Puerta de Hierro-Segovia de Arana (IDIPHISA) (PT17/0015/0020 in the Spanish National Biobanks Network), for the human specimens used in this study. This research was supported by SEOM-FONTVELLA grant and “Tu fuerza es nuestra fuerza-Galapagar” fund. 

## Figures and Tables

**Table 1 jcm-12-00065-t001:** IPF and contributing variables: factor loadings.

Variables	Factor:IPF
BMI	0.00074
Waist	1.06409
Hip	−0.26407
WHI	−0.21484
Basal heart rate	0.07361
Recovery heart rate	0.01559
Gait speed	0.08222
VO_2MAX_	−0.28282
STS	0.00212
Handgrip	−0.01497

BMI: Body Mass Index. IPF: Integrated Physical Fitness. STS: Sit-to-Stand Test. WHI: Waist-to-hip Index.

**Table 2 jcm-12-00065-t002:** Descriptive data.

	Variables	Values
	Clinical	
	Age (mean, range)	51 (34–69)
	Time since diagnosis (months, SD)	33 (21)
Stage (%)	Stage I	41
	Stage II	42
	Stage III	17
Treatment (%)	Chemotherapy	74
	Radiotherapy	62
	Anthracyclines	62
	Trastuzumab	22
	Hormonotherapy	83
	**Socio-demographics (self-reported)**	
Education level	Primary or less	9
	Secondary	30
	College	61
Employment status	Active	36
	Inactive	64
Global annual income	<EUR 10,000	21
	EUR 10–30,000	54
	EUR 30–60,000	24
	>EUR 60,000	1
	Partner (yes)	86
	Disabled family member under care	5
	Children under 15 years old	40
Perceived health	Very good	12
	Good	59
	Fair	27
	Poor	2
	Smoking	8
	**Physical condition and biomarkers**	
		Mean (SD)
Anthropometry	Height (m)	1.60 (0.07)
	Weight (kg)	69 (11.26)
	BMI (kg/m^2^)	26 (4.56)
	Waist (cm)	85 (11.42)
	Hip (cm)	103 (9.3)
	WHI	0.82 (0.06)
Physical fitness	Handgrip (kg)	24 (0.5)
	Estimated VO_2 MAX_ (mL/kg/min)	27 (7.5)
	One-mile walk test time (min)	16 (1.71)
	Basal heart rate (beats/min)	76 (12.16)
	Final heart rate (beats/min)	126 (17.35)
	Recovery heart rate (beats/min)	109 (21.32)
	Sit-to-stand test (s)	6 (2.49)
Biomarkers	CRP (mg/L)	2.1 (4.2)
	NLR	2.4 (1.5)
	Cortisol (micro g/dL)	16.7 (6.4)
	Progesterone (ng/mL)	0.80 (1.7)
	Testosterone (ng/dL)	25 (10.5)
	Androsterone	1.7 (1.9)
	Insulin (micro UI/mL)	12.5 (30.0)
	IGF-1 (ng/mL)	112.6 (55.0)
	IGFBP3 (micro g /mL)	7.7 (1.0)
	Estrogens	51.8 (90.83)

BMI: Body Mass Index. CRP: C-Reactive Protein. IGF-1: Insulin-like Growth Factor-1. IGFBP3: Insulin-like Growth Factor Binding Protein 3. NLR: Neutrophil Lymphocyte Ratio. WHI: Waist-to-Hip Index.

**Table 3 jcm-12-00065-t003:** Regression coefficients. Biomarkers and PF assessment.

	Coefficient	SE	*p*-Value
CARDIORESPIRATORY FITNESS			
**Estimated VO_2MAX_**			
CRP	−0.59	0.18	0.002 *
Insulin	−0.07	0.03	0.014 *
**Basal heart rate**			
Insulin	0.13	0.04	0.006 *
**Recovery heart rate**			
Testosterone	0.84	0.23	0.001 *
MUSCULAR COMPONENT			
**STS test**			
CRP	0.20	0.06	0.002 *
**Handgrip**			
MOTOR COMPONENT			
**Gait speed**			
CRP	0.13	0.04	0.004 *
Insulin	0.01	0.01	0.024 *
BODY COMPOSITION			
**BMI**			
CRP	0.55	0.10	0.000 *
**Hip**			
CRP	0.71	0.23	0.003 *
**Waist**			
CRP	1.04	0.28	0.000 *
IGF-1	−0.05	0.02	0.040 *
**WHI**			
CRP	0.00	0.00	0.012 *
IGF-1	−0.00	0.00	0.049 *
**IPF**			
CRP	0.09	0.02	0.000 *

* = Statistically significant. BMI: Body Mass Index. CRP: C-Reactive Protein. IGF-1: Insulin-like Growth Factor-1. IGFBP3: Insulin-like Growth Factor Binding Protein 3. IPF: Integrated Physical Fitness. NLR: Neutrophil Lymphocyte Ratio. SE: Standard error. WHI: Waist-to-Hip Index.

## Data Availability

The data presented in this study are available on request from the corresponding author.

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
