# Peer review of "C-Reactive Protein Is Associated with Physical Fitness in Breast Cancer Survivors"

_jcm, 2022, doi:10.3390/jcm12010065_

Round 1
Reviewer 1 Report
In this manuscript, María Romero-Elías et al showed that C-REACTIVE PROTEIN IS ASSOCIATED WITH PHYSICAL FITNESS IN BREAST CANCER SURVIVORS. These findings are potentially interesting but the added value of this publication lies only in the correlations between individual parameters in breast cancer survivors. It is desirable to state that this is only the first preclinical study that requires a comparison of physical fitness in patients, before treatment and monitoring during treatment and after treatment.
The manuscript could be further strengthened with a few additional edits listed below.
1. In lines 85 and 88, please correct the exponents correctly
2. In line 146, there are different percentages for hormonotherapy in the text than in table 2
3. I don't quite understand to values of biomarkers in Table 2. Why are you reporting average values? The deviations are too big, aren't they?
4. In table 3, I would suggest listing only the values where the P value is significant, the others should be listed briefly in the text, the table is unnecessarily large in this form
5. In table 3, I would suggest listing only the values where the P value is significant, the others should be listed briefly in the text, the table is unnecessarily large in this form.
6. The statement in line 283 CRP ......is a well-known risk factor for CV disease, its use for surveillance of BCS should be considered.....is unfounded because your study was conducted only on patients who survived and you do not have information about patients who succumbed to the disease. Perhaps these parameters will be similar.
Author Response
First of all, we would like to thank the editor and reviewers for providing us with positive feedback and constructive comments regarding our article, entitled C-REACTIVE PROTEIN IS ASSOCIATED WITH PHYSICAL FITNESS IN BREAST CANCER SURVIVORS.
We have addressed all the comments made by the reviewers, and we honestly feel that the quality of the article has improved as a consequence.
We provide below a detailed reply to all the reviewers’ comments. Our changes in the manuscript are highlighted in red-coloured font.
Beside that we have identified some minor mistakes, that should be corrected and can be tracked.
- We have corrected one affiliation (Héctor Cebolla)
- We propose a small but important addition to the ABSTRACT (“waist”). IGF-1 was negatively affected to the waist and waist-to-hip ratio. Also in line 180 "IGF-1 was associated with healthier waist and (-.05, p=.040) WHI”
This association was acknowledged in the table and in the discussion but was missed in the results.
RESPONSE TO REVIEWER 1
In this manuscript, María Romero-Elías et al showed that C-REACTIVE PROTEIN IS ASSOCIATED WITH PHYSICAL FITNESS IN BREAST CANCER SURVIVORS. These findings are potentially interesting, but the added value of this publication lies only in the correlations between individual parameters in breast cancer survivors. It is desirable to state that this is only the first preclinical study that requires a comparison of physical fitness in patients, before treatment and monitoring during treatment and after treatment.
The manuscript could be further strengthened with a few additional edits listed below.
- In lines 85 and 88, please correct the exponents correctly
Response 1: We have corrected the exponents in line 85 (kg/m2) and 88 (VO2MAX). Also in lines 169, 227, 234
- In line 146, there are different percentages for hormonotherapy in the text than in table 2
Response 2: Thank you very much for noticing it. There was a mistake in all the treatments in the table that thank to your comment we could identify. We have proceeded to change the percentages in the table.
- 3. I don't quite understand to values of biomarkers in Table 2. Why are you reporting average values? The deviations are too big, aren't they?
Response 3: We fully agree the comment. The wide deviations mirror the biological behavior of this kind of biomarkers. We are attaching a table with the quartiles for each biomarker that confirm the scattering of the data. However, we still think that the simplest way of informing is reporting average with SD.
If the reviewer considers that median or distribution in quartiles is more informative, we would add that information.
|
variable |
N |
mean |
sd |
p25 |
p50 |
p75 |
|
PCR |
84 |
2.202 |
4.176 |
.15 |
.7 |
2.1 |
|
NLR |
82 |
2.049 |
.9238 |
1.43 |
1.79 |
2.3 |
|
Cortisol |
84 |
16.68 |
6.391 |
11.65 |
15.2 |
20.7 |
|
Progesterone |
82 |
.8022 |
1.721 |
.19 |
.365 |
.58 |
|
Testosterone |
84 |
25 |
10.52 |
17.2 |
22.95 |
30.3 |
|
Androsterone |
82 |
1.739 |
1.885 |
.56 |
1.095 |
2.12 |
|
Insuline |
84 |
12.47 |
30.05 |
4.575 |
6.14 |
11.95 |
|
IGF1 |
84 |
112.6 |
54.97 |
74 |
95.5 |
131.5 |
|
IGFBP3 |
83 |
4.731 |
1.046 |
4 |
5 |
5 |
|
Estrogens |
84 |
51.94 |
91.37 |
14.5 |
24 |
38.5 |
- 4. In table 3, I would suggest listing only the values where the P value is significant, the others should be listed briefly in the text, the table is unnecessarily large in this form
Response 4: We do agree the reviewer’s appreciation. We have applied the comment removing all non-significant values in Table 3, with the change control.
We have added in the text “Statistically significant regression results are presented in table 3. All the biomarkers, namely CRP, NLR, cortisol, estrogens, progesterone, testosterone, androsterone, insulin, IGF-1and IGFBP3 were evaluated for each component of PF. In order to facilitate the reading, only significant results were included” (line 154- 170).
- In table 3, I would suggest listing only the values where the P value is significant, the others should be listed briefly in the text, the table is unnecessarily large in this form.
Response 5: We have replied this comment in the previous one.
- The statement in line 283 CRP ......is a well-known risk factor for CV disease, its use for surveillance of BCS should be considered.....is unfounded because your study was conducted only on patients who survived and you do not have information about patients who succumbed to the disease. Perhaps these parameters will be similar.
Response 6: Thanks for this valuable suggestion. We understand and respect the comment. And honestly we agree is not founded. We have deleted this sentence in line 283.

Reviewer 2 Report
Prolonging survival time and improving quality of life may be the best help for cancer patients at present. Therefore, it is important to find new prognostic factors that can improve the quality of life of cancer patients. CRP is a protein of the acute systemic inflammation and is, therefore, a prime marker of inflammation. Accumulating evidence suggests that C-reactive protein involved in occurring in cancer, and has been suggested as a risk factor or prognostic indicator for some tumors (PMID: 26001129, PMID: 22139643, PMID: 33933070, PMID: 34245857). Elías and Casado et al. evaluated the relationship between physical fitness and biomarkers, and found that C-reactive protein could be as a potential indicator of poor PF in BCS. Although there is a lot of papers on C-reactive protein, overall, I think their research has some scientific value. I just have four minor comments
Minor comments
1. Table 2 needs to be reformatted to make the results appear more clearly. For example, the table is designed with 3 columns, and the headings such as Stage (%), Treatment (%) and Education level et al. are combined into cells and placed in the first column.
2. The reference should be located in front of the period. Such as [13], [14], [27] et al.
3. The last part of the Discussion section should not be here. It should be placed separately in the acknowledgements section.
4. By considering the recent understanding that CRP exists in multiple isoforms with distinct biological activities (PMID: 33324413), which subtype of CRP did the authors examined?
Author Response
First of all, we would like to thank the editor and reviewers for providing us with positive feedback and constructive comments regarding our article, entitled C-REACTIVE PROTEIN IS ASSOCIATED WITH PHYSICAL FITNESS IN BREAST CANCER SURVIVORS.
We have addressed all the comments made by the reviewers, and we honestly feel that the quality of the article has improved as a consequence.
We provide below a detailed reply to all the reviewers’ comments. Our changes in the manuscript are highlighted in red-coloured font.
Beside that we have identified some minor mistakes, that should be corrected and can be tracked.
- We have corrected one affiliation (Héctor Cebolla)
- We propose a small but important addition to the ABSTRACT (“waist”). IGF-1 was negatively affected to the waist and waist-to-hip ratio. Also in line 180 "IGF-1 was associated with healthier waist and (-.05, p=.040) WHI”
This association was acknowledged in the table and in the discussion but was missed in the results.
RESPONSE TO REVIEWER 2
Prolonging survival time and improving quality of life may be the best help for cancer patients at present. Therefore, it is important to find new prognostic factors that can improve the quality of life of cancer patients. CRP is a protein of the acute systemic inflammation and is, therefore, a prime marker of inflammation. Accumulating evidence suggests that C-reactive protein involved in occurring in cancer and has been suggested as a risk factor or prognostic indicator for some tumors (PMID: 26001129, PMID: 22139643, PMID: 33933070, PMID: 34245857). Elías and Casado et al. evaluated the relationship between physical fitness and biomarkers, and found that C-reactive protein could be as a potential indicator of poor PF in BCS. Although there is a lot of papers on C-reactive protein, overall, I think their research has some scientific value. I just have four minor comments
Minor comments
1.Table 2 needs to be reformatted to make the results appear more clearly. For example, the table is designed with 3 columns, and the headings such as Stage (%), Treatment (%) and Education level et al. are combined into cells and placed in the first column.
Response 1: We understand you meant table 3 (and not 2), are we right?
We have modified the design of Table 3, according to your recommendations. Please let us know in the case we have not understood your suggestion.
The reference should be located in front of the period. Such as [13], [14], [27] et al.
Response 2: Thank you for this valuable comment. We have checked all the references accordingly.
- The last part of the Discussion section should not be here. It should be placed separately in the acknowledgements section.
Response 3: We have moved the last part of the discussion on a new section named “acknowledgements”.
- By considering the recent understanding that CRP exists in multiple isoforms with distinct biological activities (PMID: 33324413), which subtype of CRP did the authors examined?
Response 4: Thank you very much for this opportune question. This study was done in a clinical setting using biomarkers that were available in our daily practice. Therefore, we measured conventional CRP levels. We did not analyze hsCRP, monomeric or pentameric isoform.

Round 2
Reviewer 1 Report
The article in its current form is a kind of preliminary study, on which the authors should build and measure several measurements that would confirm the hypothesis they have tentatively formulated. The edits the authors made improved the article, and I have no further revision requests.